# Privacy-Aware Compression for Federated Data Analysis

**Kamalika Chaudhuri**[*1]    **Chuan Guo**[*1]    **Mike Rabbat**[1]

[1]Meta AI, USA. *Equal contribution.

## Abstract

Federated data analytics is a framework for distributed data analysis where a server compiles noisy responses from a group of distributed low-bandwidth user devices to estimate aggregate statistics. Two major challenges in this framework are privacy, since user data is often sensitive, and compression, since the user devices have low network bandwidth. Prior work has addressed these challenges separately by combining standard compression algorithms with known privacy mechanisms. In this work, we take a holistic look at the problem and design a family of privacy-aware compression mechanisms that work for any given communication budget. We first propose a mechanism for transmitting a single real number that has optimal variance under certain conditions. We then show how to extend it to metric differential privacy for location privacy use-cases, as well as vectors, for application to federated learning. Our experiments illustrate that our mechanism can lead to better utility vs. compression trade-offs for the same privacy loss in a number of settings.

## 1 INTRODUCTION

Federated data analytics is a framework for distributed data analysis and machine learning that is widely applicable to use-cases involving continuous data collection from a large number of devices. Here, a central server receives responses from a large number of distributed clients, and aggregates them to compute a global statistic or a machine learning model. An example is training and fine-tuning a speech-to-text model for a digital assistant; here a central server has a speech-to-text model, which is continuously updated based on feedback from client devices about the quality of predictions on their local data. Another example is maintaining real-time traffic statistics in a city for ride-share demand prediction; here, a central server located at the ride-share company collects and aggregates location data from a large number of user devices.

Most applications of federated data analysis involve two major challenges – privacy and compression. Since typical use-cases involve personal data from users, it is important to maintain their privacy. This is usually achieved by applying a local differentially private (LDP) algorithm (Duchi et al., 2013; Kasiviswanathan et al., 2011) on the raw inputs at the client device so that only sanitized data is transmitted to the server. Additionally, since the clients frequently have low-bandwidth high-latency uplinks, it is also important to ensure that they communicate as few bits to the server as possible. Most prior work in this area (Girgis et al., 2021; Kairouz et al., 2021; Agarwal et al., 2021) addressed these two challenges separately – first, a standard LDP algorithm is used to sanitize the client responses, and then standard compression procedures are used to compress them before transmission. However, this leads to a loss in accuracy of the client responses, ultimately leading to a loss in estimation or learning accuracy at the server. Moreover, each of these methods requires a very specific communication budget and is not readily adapted to other budgets.

In this work, we take a closer look at the problem and propose designing the privacy mechanism in conjunction with the compression procedure. To this end, we propose a formal property called *asymptotic consistency* that any private federated data analysis mechanism should possess. Asymptotic consistency requires that the aggregate statistics computed by the server converge to the non-private aggregate statistics as the number of clients grows. If the server averages the client responses, then a sufficient condition for asymptotic consistency is that the clients send an unbiased estimate of their input. Perhaps surprisingly, many existing mechanisms are not unbiased, and thus not asymptotically consistent.

We first consider designing such unbiased mechanisms that, given any communication budget $b$, transmit a continuous

*Accepted for the 38th Conference on Uncertainty in Artificial Intelligence* (UAI 2022).

scalar value that lies in the interval $[0, 1]$ with local differential privacy and no public randomness. We observe that many existing methods, such as truncated Gaussian, lead to biased solutions and asymptotically inconsistent outcomes if the inputs lie close to an end-point of the truncation interval. Motivated by this, we show how to convert two existing local differentially private mechanisms for transmitting categorical values – bit-wise randomized response (Warner, 1965) and generalized randomized response – to unbiased solutions.

We then propose a novel mechanism, the Minimum Variance Unbiased (MVU) mechanism, that given $b$ bits of communication, exploits the ordinal nature of the inputs to provide a better privacy-accuracy trade-off. We show that if the input is drawn uniformly from the set $\{0, 1/(2^b - 1), \ldots, 1\}$, then the MVU mechanism has minimum variance among all mechanisms that satisfy the local differential privacy constraints. We show how to adapt this mechanism to metric differential privacy (Andrés et al., 2013) for location privacy applications. To adapt it to differentially private SGD (DP-SGD; Abadi et al. (2016)), we then show how to extend it to vectors within an $L_p$-ball, and establish tight privacy composition guarantees.

Finally, we investigate the empirical performance of the MVU mechanism in two concrete use-cases: distributed mean estimation and private federated learning. In each case, we compare our method with several existing baselines, and show that our mechanism can achieve better utility for the same privacy guarantees. In particular, we show that the MVU mechanism can match the performance of specially-designed gradient compression schemes such as stochastic signSGD (Jin et al., 2020) for DP-SGD training of neural networks at the same communication budget.

# 2 PRELIMINARIES

In private federated data analysis, a central server calculates aggregate statistics based on sensitive inputs from $n$ clients. The statistics might be as simple as the prevalence of some event, or as complicated as a gradient to a large neural network. To preserve privacy, the clients transmit a sanitized version of their input to the server. Two popular privacy notions used for sanitization are local differential privacy (Duchi et al., 2013; Kasiviswanathan et al., 2011) and metric differential privacy (Andrés et al., 2013).

## 2.1 PRIVACY DEFINITIONS

**Definition 1.** *A randomized mechanism $\mathcal{M}$ with domain $\mathrm{dom}(\mathcal{M})$ and range $\mathrm{range}(\mathcal{M})$ is said to be $\epsilon$-local differentially private (LDP) if for all pairs $x$ and $x'$ in the domain of $\mathcal{M}$ and any $S \subseteq \mathrm{range}(\mathcal{M})$, we have that:*

$$\Pr(\mathcal{M}(x) \in S) \leq e^{\epsilon} \Pr(\mathcal{M}(x') \in S).$$

Here $\epsilon$ is a privacy parameter where lower $\epsilon$ implies better privacy. The LDP mechanism $\mathcal{M}$ is run on the client side, and the result is transmitted to the server. We assume that the clients and the server do not share any randomness. It might appear that a local DP requirement implies that a client's response contains very little useful information. While each individual response may be highly noisy, the server is still able to obtain a fairly accurate estimate of an *aggregate property* if there are enough clients. Thus, the challenge in private federated data analysis is to design protocols — privacy mechanisms for clients and aggregation algorithms for servers — so that client privacy is preserved, and the server can obtain an accurate estimate of the desired statistic.

A related definition is metric differential privacy (metric-DP) (Chatzikokolakis et al., 2013), which is also known as geo-indistinguishability (Andrés et al., 2013) and is commonly used to quantify location privacy.

**Definition 2.** *A randomized mechanism $\mathcal{M}$ with domain $\mathrm{dom}(\mathcal{M})$ and range $\mathrm{range}(\mathcal{M})$ is said to be $\epsilon$-metric DP with respect to a metric $d$ if for all pairs $x$ and $x'$ in the domain of $\mathcal{M}$ and any $S \subseteq \mathrm{range}(\mathcal{M})$, we have that:*

$$\Pr(\mathcal{M}(x) \in S) \leq e^{\epsilon d(x,x')} \Pr(\mathcal{M}(x') \in S).$$

Metric DP offers granular privacy that is quantified by the metric $d$ – inputs $x$ and $x'$ that are close in $d$ are indistinguishable, while those that are far apart in $d$ are less so.

## 2.2 PROBLEM STATEMENT

In addition to balancing privacy and accuracy, a bottleneck of federated analytics is communication since client devices typically have limited network bandwidth. Thus, the goal is to achieve privacy and accuracy along with a limited amount of communication between clients and servers. We formalize this problem as follows.

**Problem 3.** *Suppose we have $n$ clients with sensitive data $x_1, \ldots, x_n$ where each $x_i$ lies in a domain $\mathcal{X}$, and a central server $S$ seeks to approximate an aggregate statistic $\mathcal{T}_n$. Our goal is to design two algorithms, a client-side mechanism $\mathcal{M}$ and a server-side aggregation procedure $\mathcal{A}_n$, such that the following conditions hold:*

1. *$\mathcal{M}$ is $\epsilon$-local DP (or $\epsilon$-metric DP).*
2. *The output of $\mathcal{M}$ can be encoded in $b$ bits.*
3. *$\mathcal{A}_n(\mathcal{M}(x_1), \ldots, \mathcal{M}(x_n))$ is a good approximation to $\mathcal{T}_n(x_1, \ldots, x_n)$.*

Prior works addressed the communication challenge by making the clients use a standard local DP mechanism followed by a standard quantization process. We develop methods where both mechanisms are designed together so as to obtain high accuracy at the server end.

## 2.3 ASYMPTOTIC CONSISTENCY

We posit that any good federated analytics solution $(\mathcal{M}, \mathcal{A}_n)$ where $\mathcal{M}$ is a client mechanism and $\mathcal{A}_n$ is the server-side aggregation procedure should have an *asymptotic consistency* property. Loosely speaking, this property ensures that the server can approximate the target statistic $\mathcal{T}_n$ arbitrarily well with clients. Formally,

**Definition 4.** *We say that a private federated analytics protocol is* asymptotically consistent *if the output of the server's aggregation algorithm $\mathcal{A}_n(\mathcal{M}(x_1), \ldots, \mathcal{M}(x_n))$ approaches the target statistic $\mathcal{T}_n(x_1, \ldots, x_n)$ as $n \to \infty$. In other words, for any $\alpha, \delta > 0$, there exists an $n_0$ such that for all $n \geq n_0$, we have:*

$$\Pr(|\mathcal{A}_n(\mathcal{M}(x_1), \ldots, \mathcal{M}(x_n)) - \mathcal{T}_n(x_1, \ldots, x_n)| \geq \alpha) \leq \delta$$

While the server can use any aggregation protocol $\mathcal{A}_n$, the most common is a simple averaging of the client responses – $\mathcal{A}_n(\mathcal{M}(x_1), \ldots, \mathcal{M}(x_n)) = \frac{1}{n} \sum_i \mathcal{M}(x_i)$. It is easy to show the following lemma.

**Lemma 5.** *If $\mathcal{M}(x)$ is unbiased for all $x$ and has bounded variance, and if $\mathcal{A}_n$ computes the average of the client responses, then the federated analytics solution is asymptotically consistent.*

While asymptotic consistency may seem basic, it is surprisingly not satisfied by a number of simple solutions. An example is when $\mathcal{M}(x)$ is a Gaussian mechanism whose output is truncated to an interval $[a, b]$. In this case, if $x_i = a$ for all $i$, the truncated Gaussian mechanism will be biased with $\mathbb{E}[\mathcal{M}(x_i)] > x_i$, and consequently the server's aggregate will not approach $a$ for any number of clients.

Some of the recently proposed solutions for federated learning are also not guaranteed to be asymptotically consistent. Examples include the truncated Discrete Gaussian mechanism (Canonne et al., 2020; Kairouz et al., 2021) as well as the Skellam mechanism (Agarwal et al., 2021). While these mechanisms are unbiased if the range is unbounded and there are no communication constraints, their results do become biased after truncation.

## 2.4 COMPRESSION TOOL: DITHERING

A core component of our proposed mechanisms is dithering – a popular approach to quantization with a long history of use in communications (Schuchman, 1964; Gray and Stockham, 1993), signal processing (Lipshitz et al., 1992), and more recently for communication-efficient distributed learning (Alistarh et al., 2017; Shlezinger et al., 2020). Suppose our goal is to quantize a scalar value $x \in [0, 1]$ with a communication budget of $b$ bits. We consider the $B = 2^b$ points $G = \{0, \frac{1}{B-1}, \frac{2}{B-1}, \ldots, 1\}$ as the

quantization lattice; *i.e.*, the $B$ points uniformly spaced by $\Delta = 1/(B-1)$. Dithering can be seen as a random quantization function $\text{Dither} : [0, 1] \to G$ that is unbiased, *i.e.*, $\mathbb{E}[\text{Dither}(x)] = x$.[1] Moreover, the distribution of the quantization errors $\text{Dither}(x) - x$ can be made independent of the distribution of $x$.

While there are many forms of dithered quantization (Gray and Stockham, 1993), we focus on the following. If $x \in [\frac{i}{B-1}, \frac{i+1}{B-1})$ where $0 \leq i \leq B - 1$, then $\text{Dither}(x) = \frac{i}{B-1}$ with probability $(B-1)(\frac{i+1}{B-1} - x)$, and $\text{Dither}(x) = \frac{i+1}{B-1}$ with probability $(B-1)(x - \frac{i}{B-1})$. A simple calculation shows that $\mathbb{E}[\text{Dither}(x)] = x$ and moreover that the variance is bounded above by $\mathbb{E}[(\text{Dither}(x) - x)^2] \leq \Delta^2/4$. This procedure is equivalent to the *non-subtractive* dithering scheme $\text{Dither}(x) = \min_{q \in G} |q - (x - U)|$, where $U$ is uniformly distributed over the interval $[-\Delta/2, \Delta/2]$; see, e.g., (Aysal et al., 2008, Lemma 2).

## 3 SCALAR MECHANISMS

We consider Problem 3 when the input $x_i$ is a scalar in the interval $[0, 1]$, and the statistic[2] $\mathcal{T}$ is the average $\frac{1}{n} \sum_{i=1}^n x_i$. Our server side aggregation protocol will also output an average of the client responses. Our goal now is to design a client-side mechanism $\mathcal{M}$ that is $\epsilon$-local DP, unbiased, and can be encoded in $b$ bits.

**Notation.** The inputs to our client-side mechanism $\mathcal{M}$ are: a continuous value $x \in [0, 1]$, a privacy parameter $\epsilon$ and a communication budget $b$. The output is a number $i \in \{0, \ldots, B-1\}$ where $B = 2^b$, represented as a sequence of $b$ bits. Additionally, we have an alphabet $A = \{a_0, \ldots, a_{B-1}\}$ shared between the clients and server; a number $i$ transmitted by a client is decoded as the letter $a_i$ in $A$. The purpose of $A$ is to ensure unbiasedness.

### 3.1 STRATEGY OVERVIEW

Our privacy-aware compression mechanism operates in two phases. In the offline phase, it selects an input bit-width value $b_{\text{in}}$ and pre-computes an output alphabet $A$ and a sampling probability matrix $P \in \mathbb{R}^{B_{\text{in}} \times B_{\text{out}}}$, where $B_{\text{out}} = 2^b, B_{\text{in}} = 2^{b_{\text{in}}}$. Both $P$ and $A$ are shared with the server and all clients. In the online phase, the client-side mechanism $\mathcal{M}$ first uses dithering to round an input $x \in [0, 1]$ to the grid $\{0, \frac{1}{B_{\text{in}}-1}, \ldots, 1\}$ while maintaining unbiasedness, and then draws an index $j$ from the categori-

---

[1] When the number of grid points $B$ is clear from the context, we simply write $\text{Dither}(x)$ to simplify notation; otherwise we write $\text{Dither}_B(x)$ to indicate the value of $B$.

[2] To simplify notation, we drop the subscript $n$ from statistics $\mathcal{T}_n$ and aggregation functions $\mathcal{A}_n$, when the number of clients $n$ is clear from the context.

---

**Algorithm 1** Strategy for privacy-aware compression

---
1: **Input:** $x \in [0, 1]$, privacy budget $\epsilon$, communication budget $b = b_{\text{out}}$, input bit-width $b_{\text{in}}$.
2: **Offline phase:**
3: Let $B_{\text{out}} = 2^b$, $B_{\text{in}} = 2^{b_{\text{in}}}$.
4: Construct sampling probability matrix $P \in \mathbb{R}^{B_{\text{in}} \times B_{\text{out}}}$ and output alphabet $A = \{a_0, \ldots, a_{B_{\text{out}}-1}\}$ to satisfy $\epsilon$-DP and unbiasedness constraints.
5: **Online phase:**
6: $i = (B_{\text{in}} - 1) \cdot \text{Dither}(x) \in \{0, 1, \ldots, B_{\text{in}} - 1\}$.
7: Draw $j \in \{0, \ldots, B_{\text{out}} - 1\}$ from the categorical distribution defined by probability vector $P_i$.
8: **Return** $a_j$.

---

cal distribution defined by the probability vector $P_i$, where $i = (B_{\text{in}} - 1) \cdot \text{Dither}(x)$. The client then sends $a_j$ to the server. Algorithm 1 summarizes the procedure in pseudo-code. Note that the strategy generalizes to any bounded input range by scaling $x$ appropriately.

In order for $\mathcal{M}$ to satisfy $\epsilon$-DP and unbiasedness, we must impose the following constraints for the sampling probability matrix $P = [p_{i,j}]$ and output alphabet $A = \{a_j\}_{j=0}^{B_{\text{out}}-1}$:

$$\text{Row-stochasticity:} \quad \sum_{j=0}^{B_{\text{out}}-1} p_{i,j} = 1 \quad \forall i \tag{1a}$$

$$\text{Non-negativity:} \quad p_{i,j} \geq 0 \quad \forall i, j \tag{1b}$$

$$\epsilon\text{-DP:} \quad p_{i',j} e^{-\epsilon} \leq p_{i,j} \leq p_{i',j} e^{\epsilon} \quad \forall i \neq i' \tag{1c}$$

$$\text{Unbiasedness:} \quad \sum_{j=0}^{B_{\text{out}}-1} a_j p_{i,j} = \frac{i}{B_{\text{in}} - 1} \quad \forall i. \tag{1d}$$

Conditions (1a) and (1b) ensure that $P$ is a probability matrix. Condition (1c) ensures $\epsilon$-DP, while condition (1d) ensures unbiasedness. Note that these constraints only define the feasibility conditions for $P$ and $A$, and hence form the basis for a broad class of private mechanisms. In the following sections, we show that two variants of an existing local DP mechanism Randomized Response Warner (1965) – bitwise Randomized Response and generalized Randomized Response – can be realized as special cases of this family of mechanisms.

### 3.2 UNBIASED BITWISE RANDOMIZED RESPONSE

Randomized Response (RR) (Warner, 1965) is one of the simplest LDP mechanisms that sanitizes a single bit. Given a bit $y \in \{0, 1\}$, the RR mechanism outputs the $y$ with some probability $p$ and the flipped bit $1 - y$ with probability $1 - p$. If $p = \frac{1}{1 + e^{-\epsilon}}$, then the mechanism is $\epsilon$-local DP.

**Unbiased Bitwise Randomized Response Mechanism.**

The RR mechanism does not directly apply to our task as it is biased and applies to one bit. We obtain unbiasedness by using the output alphabet $A = \{-\frac{1}{e^\epsilon - 1}, \frac{e^\epsilon}{e^\epsilon - 1}\}$, and repeat the one-bit mechanism $b$ times on each bit of $x$, with a privacy budget of $\epsilon/b$ each time. It is not hard to see that unbiased RR with $b = 1$ is a special case of Algorithm 1. For $b > 1$, we can construct the resulting probability matrix $P$ by applying unbiased RR to each bit independently and similarly obtain the resulting output alphabet $A$.

We prove in Appendix A that Unbiased Bitwise Multiple RR satisfies $\epsilon$-local DP and is unbiased.

### 3.3 UNBIASED GENERALIZED RANDOMIZED RESPONSE

Generalized Randomized Response is a simple generalization of the one-bit RR mechanism for sanitizing a categorical value $x \in \{1, \ldots, K\}$. The mechanism transmits $x$ with some probability $p$, and a draw from a uniform distribution over $\{1, \ldots, K\}$ with probability $1 - p$. The mechanism satisfies $\epsilon$-local DP when $p = \frac{e^\epsilon - 1}{K + e^\epsilon - 1}$.

**Unbiased Generalized Randomized Response.** We can adapt Generalized RR to our task by dithering the input $x$ to the grid $\{0, \frac{1}{B_{\text{out}} - 1}, \ldots, 1\}$ where $B_{\text{out}} = 2^{b_{\text{out}}}$, and then transmitting the result using Generalized RR. Alternatively, we can derive the sampling probability matrix $P = \frac{e^\epsilon - 1}{B_{\text{out}} + e^\epsilon - 1} I_{B_{\text{out}}} + \frac{1}{B_{\text{out}} + e^\epsilon - 1}$, where $I_{B_{\text{out}}}$ is the identity matrix. However, this leads to a biased output. To address this, we change the alphabet to $A = \{a_0, a_1, \ldots, a_{B_{\text{out}}-1}\}$ such that unbiasedness is maintained. Specifically, for any $i \in \{0, \ldots, B_{\text{out}} - 1\}$, we need to ensure that when the input is $\frac{i}{B_{\text{out}} - 1}$, the expected output is also $\frac{i}{B_{\text{out}} - 1}$, which reduces to the following equation:

$$a_i \cdot \frac{e^\epsilon - 1}{B_{\text{out}} + e^\epsilon - 1} + \sum_{j=0}^{B_{\text{out}}-1} a_j \cdot \frac{1}{B_{\text{out}} + e^\epsilon - 1} = \frac{i}{B_{\text{out}} - 1}.$$

Writing this down for each $i$ gives $B_{\text{out}}$ linear equations, solving which will give us the values of $a_0, \ldots, a_{B_{\text{out}}-1}$. We establish the privacy and unbiasedness properties of Unbiased Generalized RR in Appendix A. A similar unbiased adaptation was also considered by Balle et al. (2019).

### 3.4 THE MVU MECHANISM

A challenge with Unbiased Bitwise RR and Unbiased Generalized RR is that both algorithms are not intrinsically designed for ordinal or numerical values, which may result in poor accuracy upon aggregation. We next propose a new method that improves estimation accuracy by reducing the variance of each client's output while retaining unbiasedness and hence asymptotic consistency.

Our proposed method – the *Minimum Variance Unbiased*

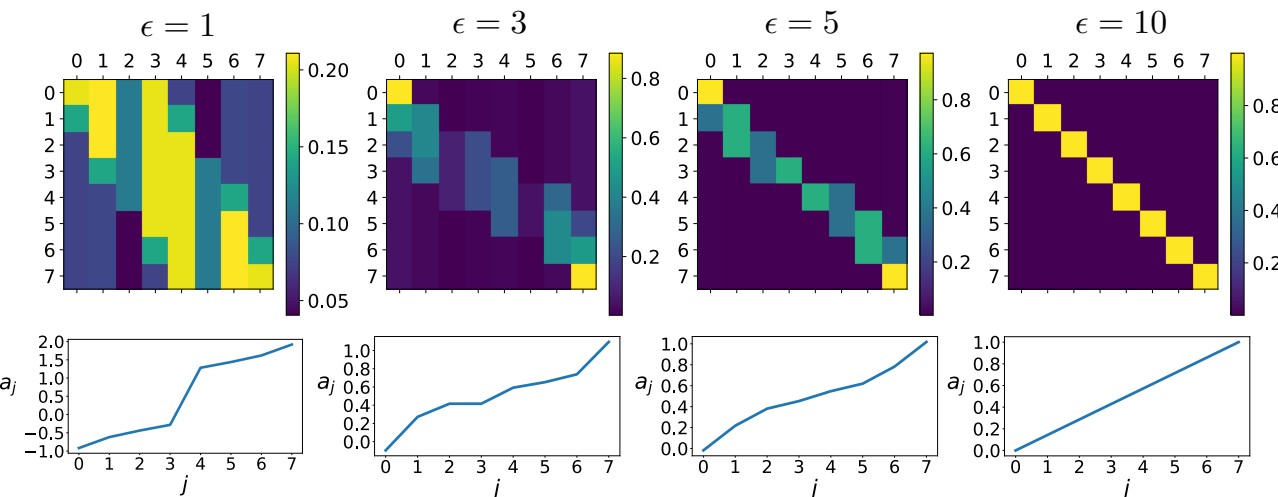

Figure 1: Optimized sampling probability matrix $P$ (top row) and output alphabet $A = \{a_0, \dots, a_{B_{out}-1}\}$ (bottom row) of the MVU mechanism with $b_{in} = b_{out} = 3$ for $\epsilon = 1, 3, 5, 10$. At $\epsilon = 1$, the DP constraint forces entries in each column to be similar, and the unbiasedness constraint causes the magnitude of $a_j$ to be large. At $\epsilon = 10$, the weaker DP constraint allows the optimal $P$ matrix to become close to the identity matrix and $a_j \approx j/(B-1)$.

(MVU) mechanism – addresses this problem by directly minimizing the variance of the client's output. This is done by solving the following optimization problem:

$$\min_{\substack{p \in [0,1]^{B_{in} \times B_{out}} \\ a \in \mathbb{R}^{B_{out}}}} \sum_{i=0}^{B_{in}-1} \sum_{j=0}^{B_{out}-1} p_{i,j} \left( \frac{i}{B_{in}-1} - a_j \right)^2 \quad (2)$$

$$\text{subject to} \quad \text{Conditions (1a)} - \text{(1d)}.$$

The objective in (2) measures the variance of the output of the mechanism when the input $i$ is uniformly distributed over the set $\{0, \frac{1}{B_{in}-1}, \dots, 1\}$. Conditions (1a)-(1d) ensure that the MVU mechanism is $\epsilon$-DP and unbiased, hence satisfying requirements for our task.

**Solving the MVU mechanism design problem.** We solve (2) using one of two approaches depending on size of the probability matrix $P$ and $\epsilon$. For smaller problems and when $\epsilon$ is not too small, we use a trust region interior-point solver (Conn et al., 2000). As $\epsilon$ approaches 0, the problem becomes poorly conditioned and we only approximately solve the problem by relaxing the unbiasedness constraint (1d). In this case we use an alternating minimization heuristic where we alternate between fixing the values $a_j$ and solving for $p_{i,j}$, and holding $p_{i,j}$ fixed and solving for $a_j$, while incorporating constraint (1d) as a soft penalty in the objective. Each of the corresponding subproblems is a quadratic program and can be solved efficiently. Figure 1 shows examples of the MVU mechanism for $b_{in} = b_{out} = 3$ and $\epsilon \in \{1, 3, 5, 10\}$ obtained using the trust region solver.

**Relationship between DP and compression.** The MVU mechanism highlights an intriguing connection between DP and compression. Since the mechanism hides information

in the input $x$ by perturbing it with random noise, as $\epsilon \to 0$, fewer bits are required to describe the noisy output $\mathcal{M}(x)$. In the limiting case of $\epsilon = 0$, all information is lost and the output can be described by zero bits. In Appendix B, we demonstrate this argument concretely by showing that as $\epsilon \to 0$, the marginal benefit of having a larger communication budget decreases.

## 4 EXTENSIONS

We now show how to extend the MVU mechanism to obtain privacy-aware and accurate compression mechanisms for metric-DP and vector spaces.

### 4.1 METRIC DP

In location privacy, client devices send their obscured locations to a central server for aggregation. Metric DP (Definition 2) is a variation of LDP that applies to this use-case. We are given a position $x$ and a metric $d$ which measures how far apart two positions are. Our goal is to output a private position $x'$ so that fine-grained properties of $x$ (such as, exact address, city block) are hidden, while coarse-grained properties (such as, city, or zip-code) are preserved.

We show how to adapt the MVU mechanism to metric DP. For simplicity, suppose that we measure position on the line, so $x \in [0, 1]$. We modify Condition (1c) to instead satisfy the metric DP constraint with respect to the metric $d$:

$$p_{i',j} e^{-\epsilon d(i/(B_{in}-1), i'/(B_{in}-1))} \le p_{i,j} \le p_{i',j} e^{\epsilon d(i/(B_{in}-1), i'/(B_{in}-1))}.$$
$$(3)$$

Thus we can get an MVU mechanism for metric DP by solv-

ing the modified optimization problem in (2) and following the same procedure in Algorithm 1.

## 4.2 EXTENSION TO VECTOR SPACES

We next look at extending the MVU mechanism to vector spaces. Specifically, a client now holds a $d$-dimensional vector $\mathbf{x}$ in a domain $\mathcal{X} \subseteq \mathbb{R}^d$, and its goal is to output an $\epsilon$-local DP version that can be communicated in $bd$ bits. The domain $\mathcal{X}$ is typically a unit $L_p$-norm ball for $p \geq 1$.

A plausible approach is to apply the scalar MVU mechanism independently for each coordinate of $\mathbf{x}$. While this will provide the optimal accuracy for $p = \infty$, for $p < \infty$, the client's variance will be higher. A second approach is to extend the MVU mechanism directly to $\mathcal{X}$ by using an alphabet $A \times A \times \ldots \times A = A^d$ and then solving the corresponding optimization problem (2). Unfortunately this is computationally intractable even for moderate $d$.

Instead, we show how to obtain a more computationally tractable approximation when $\mathcal{X}$ is an $L_p$-ball. We are motivated by the following lemma.

**Lemma 6.** *Let $\mathcal{X}$ be the unit $L_p$-ball with diameter $\Delta$. Suppose $\mathcal{M}$ is an $\epsilon$-metric DP scalar mechanism with $d(y, y') = |y - y'|^p$. Then, the mechanism $\mathcal{M}_d : \mathcal{X} \to \mathbb{R}^d$ that maps $\mathbf{x}$ to the vector $(\mathcal{M}(\mathbf{x}_1), \ldots, \mathcal{M}(\mathbf{x}_d))$ is $\epsilon\Delta^p$-local DP. Additionally, if $\mathcal{M}$ is unbiased, then $\mathcal{M}_d$ is unbiased as well.*

Lemma 6 suggests the following algorithm: Use the MVU mechanism for $\epsilon$-metric DP with $d(y, y') = |y - y'|^p$ for each coordinate, then combine to get an $\epsilon$-local DP solution for vectors with $L_p$-sensitivity $\Delta$. Since $\|\cdot\|_\infty \leq \|\cdot\|_p$, each coordinate of $\mathbf{x}$ lies in a bounded range $[-\Delta, \Delta]$, so we can scale $\mathbf{x}$ by $\mathbf{x}' \leftarrow (\mathbf{x} + \Delta)/2\Delta$ so that all entries belong to $[0, 1]$ and the MVU mechanism can be applied to $\mathbf{x}'$. Note that this scaling operation changes the $L_p$-sensitivity to $1/2$.

This solution is computationally tractable since we only need to solve an optimization problem for the scalar MVU mechanism – so involving $\approx B_{\text{out}}^2 = 2^{2b_{\text{out}}}$ variables and constraints (instead of $\approx 2^{2b_{\text{out}}d}$). We investigate how this mechanism works in practice in Section 5.

## 4.3 COMPOSITION USING RÉNYI-DP

Repeated applications of the MVU mechanism will give an additive sequential privacy composition guarantee as in standard $\epsilon$-DP. We next show how to get tighter composition bounds for the MVU mechanism using RDP accounting as in Mironov (2017).

Suppose that $\mathbf{x}, \mathbf{x}' \in \{0, 1/(B_{\text{in}}-1), \ldots, 1\}^d$ are quantized $d$-dimensional vectors, and let $Q_0, Q_1$ be the output distributions of the mechanism $\mathcal{M}$ for inputs $\mathbf{x}, \mathbf{x}'$, respectively.

By the definition of Rényi divergence (Rényi, 1961),

$$D_\alpha(Q_0 \| Q_1) = \frac{1}{\alpha - 1} \sum_{l=1}^{d} \log \sum_{j=0}^{B_{\text{in}}-1} \frac{p_{\mathbf{i}_l,j}^\alpha}{p_{\mathbf{i}'_l,j}^{\alpha-1}},$$

where $\mathbf{i}, \mathbf{i}' \in \{0, 1, \ldots, B_{\text{in}}-1\}^d$ are such that $\mathbf{x} = \mathbf{i}/(B_{\text{in}}-1)$ and $\mathbf{x}' = \mathbf{i}'/(B_{\text{in}}-1)$. Let $D^\alpha$ denote the $B_{\text{in}} \times B_{\text{in}}$ matrix with entries $D_{i,i'}^\alpha = \frac{1}{\alpha-1} \log \sum_{j=0}^{B_{\text{in}}-1} p_{i,j}^\alpha / p_{i',j}^{\alpha-1}$. Then, computation of the $\alpha$-RDP parameter for $\mathcal{M}$ can be formulated as the following combinatorial optimization problem:

$$\max_{\mathbf{i}, \mathbf{i}' \in \{0,1,\ldots,B_{\text{in}}-1\}^d} \sum_{l=1}^{d} D_{\mathbf{i}_l, \mathbf{i}'_l}^\alpha \quad \text{s.t. } \|\mathbf{i} - \mathbf{i}'\|_p^p \leq (B_{\text{in}}-1)^p \Delta^p.$$

This optimization problem is in fact an instance of the *multiple-choice knapsack problem* (Sinha and Zoltners, 1979) and admits an efficient linear program relaxation by converting the integer vectors $\mathbf{i}, \mathbf{i}'$ to probability vectors, *i.e.*,

$$\max_{\mathbf{p} \in \mathbb{R}^{d \times B_{\text{in}} \times B_{\text{in}}}} \quad \sum_{l=1}^{d} \langle D^\alpha, \mathbf{p}_l \rangle_F \tag{4}$$

$$\text{subject to} \quad \sum_{l=1}^{d} \langle C, \mathbf{p}_l \rangle_F \leq (B_{\text{in}} - 1)^p \Delta^p$$

$$\sum_{i,j} (\mathbf{p}_l)_{ij} \leq 1 \text{ and } \mathbf{p}_l \geq 0 \; \forall l,$$

where $\langle \cdot, \cdot \rangle_F$ denotes Frobenius (vectorized) inner product and $C$ denotes the distance matrix with entries $C_{ij} = (i - j)^p$. This LP relaxation can still be prohibitively expensive to solve for large $d$ since $\mathbf{p}$ contains $dB_{\text{in}}^2$ variables. Fortunately, in such cases, we can obtain an upper bound via the greedy solution; see Appendix A for the proof.

**Lemma 7.** *Let $(i^*, j^*) = \arg\max_{i,j} D_{ij}^\alpha / C_{ij}$ and let $d_0 = (B_{in} - 1)^p \Delta^p / C_{i^* j^*}$. Then (4) $\leq d_0 D_{i^* j^*}^\alpha$.*

To summarize, for composition with RDP accounting at order $\alpha$, we can either solve the LP relaxation in (4) or compute the greedy solution to obtain an upper bound for $D_\alpha(P \| Q)$, and then apply the usual composition for RDP.

## 5 EXPERIMENTS

We evaluate the MVU mechanism on two sets of experiments: Distributed mean estimation and federated learning. Our goal is to demonstrate that MVU can attain a better privacy-utility trade-off at low communication budgets compared to other private compression mechanisms. Code to reproduce our results can be found in the repo `https://github.com/facebookresearch/dp_compression`.

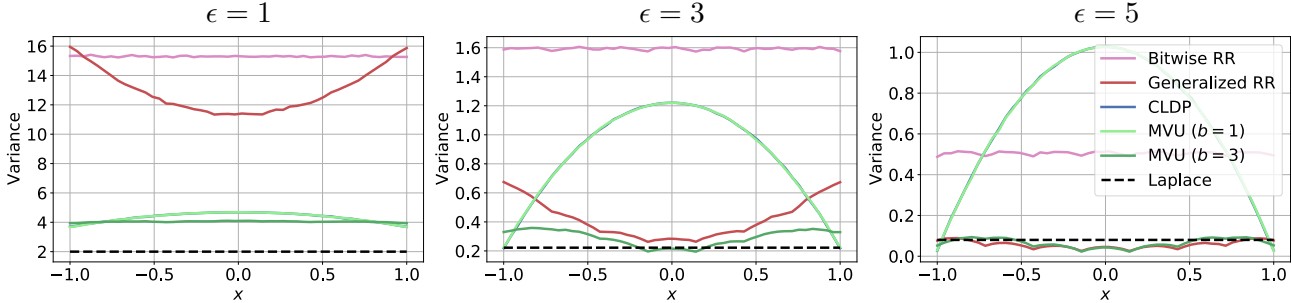

Figure 2: Distributed mean estimation for scalar data with LDP $\epsilon = 1, 3, 5$. The MVU mechanism with budget $b = 1$ recovers the CLDP mechanism and the two curves coincide, while with $b = 3$ MVU attains a low variance across all input values compared to the baseline mechanisms. See text for details.

## 5.1 DISTRIBUTED MEAN ESTIMATION

In distributed mean estimation (DME), a set of $n$ clients each holds a private vector $\mathbf{x}_i \in \mathbb{R}^d$, and the server would like to privately estimate the mean $\bar{\mathbf{x}} = \frac{1}{n} \sum_{i=1}^{n} \mathbf{x}_i$.

**Scalar DME.** We first consider the setting of scalar data, *i.e.*, $d = 1$. For a fixed value $x \in [-1, 1]$, we set $\mathbf{x}_i = x$ for all $i = 1, \ldots, n$ with $n = 100,000$ and then privatize them before taking average. We measure the squared difference between the private estimate and $\bar{\mathbf{x}} = x$, which is coincidentally the variance of the mechanism at $x$. The baseline mechanisms that we evaluate against are (unbiased) Bitwise Randomized Response (bRR), (unbiased) Generalized Randomized Response (gRR), the communication-limited local differentially private (CLDP) mechanism (Girgis et al., 2021), and the Laplace mechanism without any compression. The CLDP mechanism uses a fixed communication budget of $b = 1$, whereas for bRR and gRR we set $b = 3$, and for MVU we set $b = 1, 3$.

Figure 2 shows the plot of input value $x$ vs. variance of the private mechanism at $x$. Interestingly, MVU with $b = 1$ recovers the CLDP mechanism for $\epsilon = 1, 3, 5$, while MVU with $b = 3$ is consistently the lowest variance private compression mechanism. For larger $\epsilon$, it is evident that the variance of both gRR and MVU are comparable or even slightly lower that of the Laplace mechanism, even when compressing to only $b = 3$ bits in their output.

**Vector DME.** We next look at vector data with $d = 128$ and $n = 10,000$. We draw the sensitive vectors from two distinct distributions[3]: (i) Uniform at random from $[0, 1]^d$ and then normalize to $L_1$-norm of 1; and (ii) Uniform over the spherical sector $\mathbb{S}^{d-1} \cap \mathbb{R}_{\geq 0}^d$. In these settings, the vectors $\mathbf{x}_i$ have $L_1$- and $L_2$-sensitivity of 1, respectively.

For baselines, we consider the CLDP mechanism (Girgis et al., 2021), the Skellam mechanism (Agarwal et al., 2021),

the Laplace mechanism (for setting (i) only), and the Gaussian mechanism (for setting (ii) only). Both the Skellam and the Gaussian mechanisms are $(\epsilon, \delta)$-DP for $\delta > 0$. For a given $\epsilon > 0$, we set $\delta = 1/(n + 1)$ and choose the noise parameter $\mu$ for the Skellam mechanism using the optimal RDP conversion, and the noise parameter $\sigma$ for the Gaussian mechanism using the analytical conversion in Balle and Wang (2018). For communication budget, we set $b = 3$ for MVU and $b = 16$ for Skellam (which requires a large $b$ in order to prevent truncation error). The CLDP mechanism does not allow flexible selection of communication budget, and instead outputs a *total* number of $\log_2(d) + 1$ bits for the $L_1$-sensitivity setting, and $b = \log_2(d) + 1 = 8$ bits *per coordinate* for the $L_2$-sensitivity setting. See Appendix B for a more detailed explanation.

Figure 3 shows the mean squared error (MSE) for privately estimating $\bar{\mathbf{x}}$ across different values of $\epsilon$. In the left plot corresponding to the $L_1$-sensitivity setting, MVU can attain MSE close to the Laplace mechanism at a greatly reduced $b = 3$ bits per coordinate. In comparison, CLDP and Skellam attain MSE that is more than an order of magnitude higher than Laplace.

The right plot corresponds to $L_2$-sensitivity. Here, the MVU mechanism (dark green line) is significantly less competitive than the baselines. This is because the $L_2$-metric DP constraint for the MVU mechanism forces rows of the sampling probability matrix $P$ to be near-identical, hence is near-singular and does not admit a well-conditioned unbiased solution. To address this problem, we instead optimize the MVU mechanism to satisfy $L_1$-metric DP and use the Rényi accounting in Section 4.3 to compute its RDP guarantee, then apply RDP-to-DP conversion to give an $(\epsilon, \delta)$-DP guarantee at $\delta = \frac{1}{n+1}$. The light green line shows the performance of the $L_1$-metric DP mechanism, which now slightly outperforms both CLDP and Skellam at a much lower communication budget of $b = 3$. These results demonstrate that the MVU mechanism attains better utility vs. compression trade-off for vector data as well.

---

[3]We intentionally avoided zero-mean distributions since some of the private mechanisms converge to the all-zero vector as $\epsilon \to 0$.

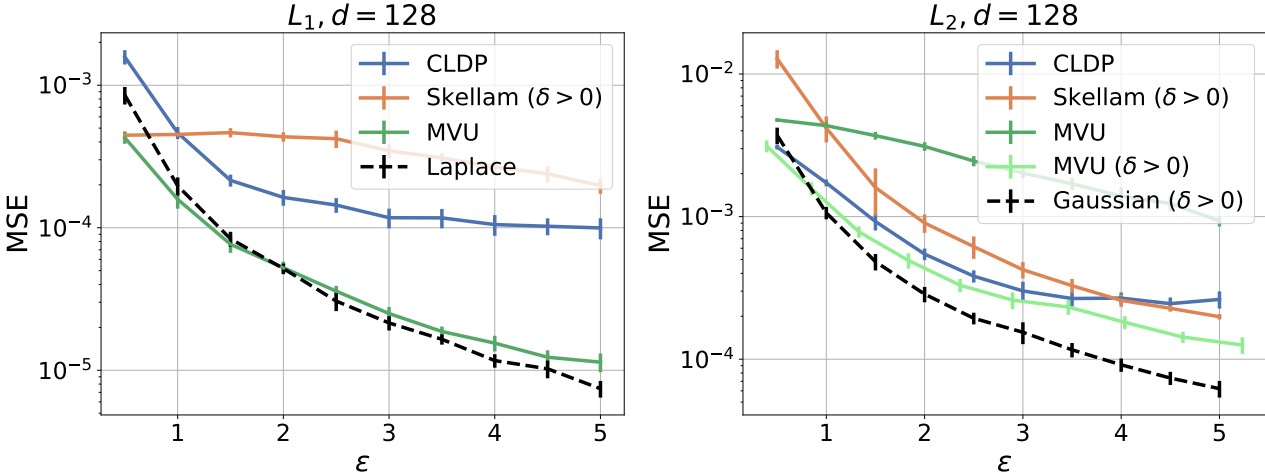

Figure 3: Distributed mean estimation for $n = 10,000$ data vectors with $L_1$- (left) and $L_2$-sensitivity (right). Error bars represent standard deviation across 10 repeated runs with different private vectors. Methods that are $(\epsilon, \delta)$-DP use the same value of $\delta = 1/(n+1)$. The MVU mechanism can attain an MSE close to that of the Laplace and Gaussian mechanisms while compressing the output to only $b = 3$ bits per coordinate.

## 5.2 PRIVATE SGD

Federated learning (McMahan et al., 2017) often employs DP to protect the privacy of the clients' updates. We next evaluate the MVU mechanism for this use case and show that it can serve as a drop-in replacement for the Gaussian mechanism for FL protocols, providing similar DP guarantees for the client update while reducing communication.

In detail, for MNIST and CIFAR-10 (Krizhevsky et al., 2009), we train a linear classifier on top of features extracted by a scattering network (Oyallon and Mallat, 2015) similar to the one used in Tramer and Boneh (2020); see Appendix B for details. The base private learning algorithm is DP-SGD with Gaussian gradient perturbation (Abadi et al., 2016) and Rényi-DP accounting. The private compression baselines are the MVU mechanism with budget $b = 1$ and stochastic signSGD (Jin et al., 2020) – a specialized private gradient compression scheme for federated SGD that applies the Gaussian mechanism and outputs its coordinate-wise sign. Similar to the distributed mean estimation experiment with $L_2$-sensitivity, we optimize the MVU mechanism to satisfy $L_1$-metric DP and then compute its Rényi privacy guarantee as in Section 4.3.

Figure 4 shows the privacy-utility trade-off curves. We sweep over a grid of hyperparameters (see Appendix B for details) for each mechanism and plot the resulting $\epsilon$ and test accuracy as a point in the scatter plot. The dashed line is the Pareto frontier of optimal privacy-utility trade-off. The result shows that MVU mechanism outperforms signSGD— a specially-designed gradient compression mechanism for federated learning—at nearly all privacy budgets with the same communication cost of *one bit per coordinate*. We include an additional result for a small convolutional network

in Appendix B, where we observe similar findings.

## 6 RELATED WORK

Federated data analysis with local DP is now a standard solution for analyzing sensitive data held by many user devices. A body of work (Erlingsson et al., 2014; Kairouz et al., 2016; Acharya et al., 2019) provides methods for analytics over categorical data. The main methods here are Randomized Response (Warner, 1965), RAPPOR (Erlingsson et al., 2014) and the Hadamard Mechanism (Acharya et al., 2019). Chen et al. (2020) shows that the Hadamard Mechanism uses near-optimal communication for categorical data.

In work on federated statistics or learning for real-valued data, Cormode and Markov (2021) provides asymptotically consistent algorithms for transmitting scalars. They propose to first sample one or a subset of indices of bits in the fixed-point representation of the input, and then apply randomized response independently to each of these bits. Girgis et al. (2020) provides mechanisms for distributed mean estimation from vectors inside unit $L_p$ balls. Unlike our method, which provides a near-optimal solution under any given communication budget, their methods use specific communication budgets and are not readily generalizable to any budget $b$. Finally, Amiri et al. (2021) propose to obtain a quantized DP mechanism by composing subtractive dithering with the Gaussian mechanism, and doing privacy accounting that factors in both. In contrast, we simply use (non-subtractive) dithering to initially obtain a fixed-point representation, and then design a mechanism to quantize and provide DP.

A large body of work focuses on federated optimization methods with compressed communication (Konečný et al.,

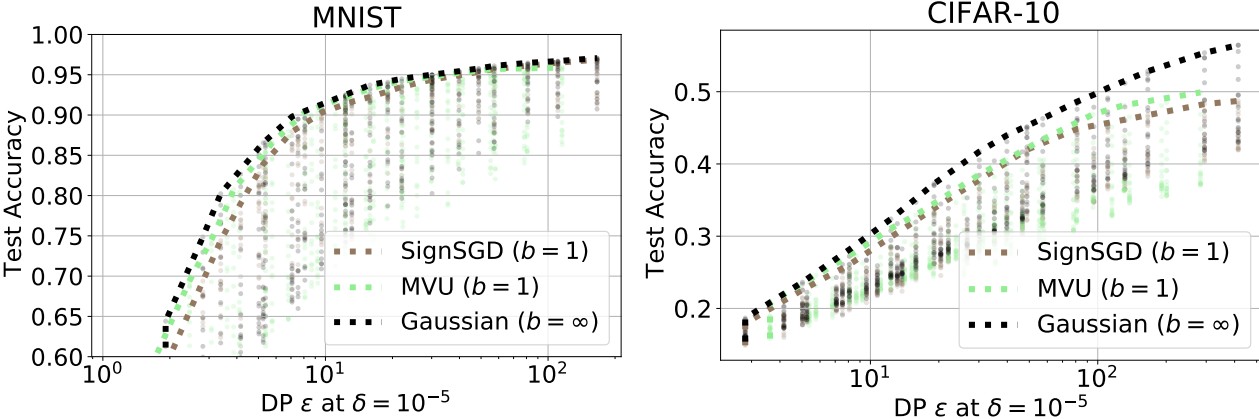

Figure 4: DP-SGD training with Gaussian mechanism, stochastic signSGD and MVU mechanism on MNIST (left) and CIFAR-10 (right). Each point corresponds to a single hyperparameter setting, and dashed line shows Pareto frontier of privacy-utility trade-off. MVU mechanism outperforms signSGD at the same communication budget of $b = 1$.

2016; Horváth et al., 2019; Das et al., 2020; Haddadpour et al., 2021; Gorbunov et al., 2021). While most propose biased compression methods (e.g., top-$k$ sparsification), such approaches require the use of error feedback to avoid compounding errors (Seide et al., 2014; Stich and Karimireddy, 2020). However, error feedback is inherently incompatible with DP (Jin et al., 2020), unlike our MVU mechanism.

## 7 CONCLUSION AND LIMITATIONS

We introduce the MVU framework to jointly design scalar compression and DP mechanisms, and extend it to the vector and metric-DP settings. We show that the MVU mechanism attains a better utility-compression trade-off for both scalar and vector mean estimation compared to other approaches in the literature. Our work shows that co-designing the compression and privacy-preserving components can lead to more efficient differentially private mechanisms for federated data analysis.

**Limitations.** Our work presents several opportunities for further improvement. 1. For vector dithering, Appendix B shows that the input vector's norm can increase by a small additive factor. Our current solution of conditional random dithering introduces a small but non-negligible bias. Future work on unbiased norm-preserving vector dithering may be able to alleviate this issue. 2. Optimizing the MVU mechanism for large values of the input/output bit width $b_{\text{in}}$ and $b_{\text{out}}$ can be prohibitively expensive, even with the alternating minimization heuristic. In order to scale the solution to higher-dimensional vectors, further effort in designing more efficient solutions for the MVU mechanism may be needed. 3. While our work focuses on local differential privacy, it may be possible to combine our approach with secure aggregation protocols to derive central differential privacy guarantees. However, since the MVU mechanism is not additive, further analysis is required to characterize the distribution of the aggregate for our mechanism, which we leave for future work.

## ACKNOWLEDGEMENTS

We thank Graham Cormode, Huanyu Zhang, and anonymous reviewers for insightful comments and suggestions that helped shape our final draft.

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
