# OpenReview forum: "Privacy-Aware Compression for Federated Data Analysis"
_auai.org/UAI/2022/Conference — UAI 2022 Poster_

### Official Review · Reviewer_sS2Z · 2022-04-13

**Q2(1) Originality/Novelty:** 3
**Q2(2) Significance/Impact:** 3
**Q2(3) Correctness/Technical Quality:** 3
**Q2(6) Clarity Of Writing:** 3
**Q6 Overall Score:** 6
**Q8 Confidence In Your Score:** 3

**Q1 Summary And Contributions:**

1. propose asymptotic consistency property to consider privacy and compression together
2. convert two existing local differentially private mechanisms for transmitting categorical values to unbiased solutions
3. propose the Minimum Variance Unbiased (MVU) mechanism to get better balance between privacy and accuracy
4. validate the performance on two concrete use-cases and show better utility for the same privacy guarantees

**Q2 Assessment Of The Paper:**

More detailed information regarding each of these aspects is given below:

**Q2(4) Quality Of Experiments (Optional):**

3: Good: The experimental evaluation is adequate, and the results convincingly support the main claims.

**Q2(5) Reproducibility:**

3: Good: Key resources (e.g., proofs, code, data) are available and key details (e.g., proofs, experimental setup) are sufficiently well-described for competent researchers to confidently reproduce the main results.

**Q3 Main Strengths:**

1. This paper provides a new perspective of solving the problem of privacy and compression in federated data analysis task and shows that solving these two challenges jointly will result in more accurate models than solving them with separate methods.
2. The proposed methods attains better utility-compression trade-off for both scalar and vector mean estimation compared to prior works.
3. Detailed theory analysis of the proposed algorithm is provided and the experimental results validate effectiveness of proposed methods.

**Q4 Main Weakness:**

One major question I have is that the experiments are only done on the MNIST dataset, which may not be enough. It is recommended to validate the methods on other datasets, for example CIFAR-10 or Tiny ImageNet datasets, as the paper ‘Scalable Differential Privacy with Certified Robustness in Adversarial Learning’ (ICML 2020) has done.

Besides, the data setting in federated learning like the number of clients, the local epochs, the data distribution among clients and their effects on the proposed methods are not stated clearly.

**Q5 Detailed Comments To The Authors:**

1. From the perspective of federated learning, it is not clear that whether the reported performance of the methods is in iid or non-iid data settings. The client setting in the federated learning experiments is recommended to be illustrated in detail, for example, the number of clients, the data distribution on each client, etc.
2. Are the methods sensitive to the hypermeters in federated learning, for example the local update epochs?

**Q7 Justification For Your Score:**

This papers attempts to balance the privacy and accuracy in federated learning. Instead of combing existing privacy-aware tools and compression methods with existing federated learning algorithms, this paper provides a unified framework and solve these two challenges jointly. The perspective of the proposed methods is novel in this research field. Besides, the proof of the theory and experiments validate the effectiveness of the proposed methods.

**Q9 Complying With Reviewing Instructions:**

1: Yes.

---

### Official Review · Reviewer_h4tR · 2022-04-16

**Q2(1) Originality/Novelty:** 3
**Q2(2) Significance/Impact:** 3
**Q2(3) Correctness/Technical Quality:** 3
**Q2(6) Clarity Of Writing:** 3
**Q6 Overall Score:** 5
**Q8 Confidence In Your Score:** 4

**Q1 Summary And Contributions:**

The paper considers an ε-local differentially mechanism with inputs from domain [0, 1] and with a fixed number of bits required to encode its output, such that average of the outputs is a good approximation of average of the inputs.


**Q2 Assessment Of The Paper:**

More detailed information regarding each of these aspects is given below:

**Q2(4) Quality Of Experiments (Optional):**

3: Good: The experimental evaluation is adequate, and the results convincingly support the main claims.

**Q2(5) Reproducibility:**

3: Good: Key resources (e.g., proofs, code, data) are available and key details (e.g., proofs, experimental setup) are sufficiently well-described for competent researchers to confidently reproduce the main results.

**Q3 Main Strengths:**

The mechanism uses the following components:
-- Dithering to discretize continuous inputs
-- An alphabet a_0, …, a_{B-1} and a matrix P such that for input i the output is a_j with Probability P_ij. They must satisfy certain constraints to ensure unbiasedness, ε-DP and probability constraints.
-- Under these constraints, the mechanism minimizes variance

Compared with existing work, the paper doesn’t perform input sanitization and compression separately, but instead devises a mechanism that naturally combines these tasks.

**Q4 Main Weakness:**

I have the following concerns about the paper:
-- I didn’t fully understand the case of small ε:
1) How exactly relaxing (1d) helps to handle this case?
2)  While Page 5 discusses how to handle small ε, the experiments only consider large values of ε. Do I understand correctly that the -- algorithm performs worse than other alternatives for small ε? This looks like a serious limitation of the algorithm




**Q5 Detailed Comments To The Authors:**

-- Experiments: I believe that it’s not sufficient to consider MNIST for federated learning experiments
-- In Equation (2), you assume that all inputs are uniformly distributed. However, this is not necessarily the case: e.g. in the federated learning, it’s likely that the gradients don’t significantly change between iterations

**Q7 Justification For Your Score:**

The paper has novel contributions and can be considered for publication, although the contributions don't appear to be particularly strong.

**Q9 Complying With Reviewing Instructions:**

1: Yes.

---

### Official Review · Reviewer_JsX2 · 2022-04-18

**Q2(1) Originality/Novelty:** 4
**Q2(2) Significance/Impact:** 4
**Q2(3) Correctness/Technical Quality:** 3
**Q2(6) Clarity Of Writing:** 4
**Q6 Overall Score:** 8
**Q8 Confidence In Your Score:** 3

**Q1 Summary And Contributions:**

This paper looks at federated data analysis through
a lens that combines both privacy concerns with
compression flexibility.


**Q2 Assessment Of The Paper:**

More detailed information regarding each of these aspects is given below:

**Q2(5) Reproducibility:**

3: Good: Key resources (e.g., proofs, code, data) are available and key details (e.g., proofs, experimental setup) are sufficiently well-described for competent researchers to confidently reproduce the main results.

**Q3 Main Strengths:**

This paper has fundamentally changed the way that I look at federated learning.
Looking back on it, I don't understand why nobody thought of doing this before.

I find especially interesting the revelation that
local DP mechanisms can be biased - that hadn't
occurred to me either.



**Q4 Main Weakness:**

The five extra pages of supplementary material, consisting
of both proofs and extra analysis make me wonder whether
an 8-page conference format is the right one for this
work.  This belongs in a good journal.


**Q5 Detailed Comments To The Authors:**

p. 1: "If the server averages the client responses" - awkward, rephrase

**Q7 Justification For Your Score:**

My only hesitation is that it would benefit from a forum for larger contributions.

**Q9 Complying With Reviewing Instructions:**

1: Yes.

---

### Decision · Program_Chairs · 2022-05-15

**Decision:**

Accept (Poster)

**Comment:**

Meta Review: This paper designs a family of privacy-aware compression mechanisms for federated data analysis.

Quality: The paper contains sound theoretical results. The experiments can be improved.

Clarity: It is a well-written paper.

Originality: The work is original. I quote a reviewer's comment below.

“This paper has fundamentally changed the way that I look at federated learning. Looking back on it, I don't understand why nobody thought of doing this before.

I find especially interesting the revelation that local DP mechanisms can be biased - that hadn't occurred to me either.”

Significance: The results change the fundamental view of federated learning.

All reviewers are positive about the paper. The idea is original, and the analyses are sound. Some minor concerns have been raised for experiments, and the authors have updated the experimental results.